# Influence of Selective Laser Melting Technological Parameters on the Mechanical Properties of Additively Manufactured Elements Using 316L Austenitic Steel

**DOI:** 10.3390/ma13061449

**Published:** 2020-03-22

**Authors:** Janusz Kluczyński, Lucjan Śnieżek, Krzysztof Grzelak, Jacek Janiszewski, Paweł Płatek, Janusz Torzewski, Ireneusz Szachogłuchowicz, Krzysztof Gocman

**Affiliations:** 1Military University of Technology, Faculty of Mechanical Engineering, Institute of Robots & Machine Design, 2 Gen. S. Kaliskiego Street, 00-908 Warsaw 49, Poland; lucjan.sniezek@wat.edu.pl (L.Ś.); krzysztof.grzelak@wat.edu.pl (K.G.); janusz.torzewski@wat.edu.pl (J.T.); ireneusz.szachogluchowicz@wat.edu.pl (I.S.); 2Military University of Technology, Faculty of Mechatronics and Aviation, Institute of Armament Technology, 2 Gen. S. Kaliskiego Street, 00-908 Warsaw 49, Poland; jacek.janiszewski@wat.edu.pl (J.J.); pawel.platek@wat.edu.pl (P.P.); 3Military University of Technology, Faculty of Mechanical Engineering, Institute of Vehicles & Transportation, 2 Gen. S. Kaliskiego Street, 00-908 Warsaw 49, Poland; krzysztof.gocman@wat.edu.pl

**Keywords:** 316L austenitic steel, selective laser melting, powder bed fusion, technological parameters, mechanical property characterization

## Abstract

The main aim of this study was to investigate the influence of different energy density values used for the additively manufactured elements using selective laser melting (SLM).The group of process parameters considered was selected from the first-stage parameters identified in preliminary research. Samples manufactured using three different sets of parameter values were subjected to static tensile and compression tests. The samples were also subjected to dynamic Split–Hopkinson tests. To verify the microstructural changes after the dynamic tests, microstructural analyses were conducted. Additionally, the element deformation during the tensile tests was analyzed using digital image correlation (DIC). To analyze the influence of the selected parameters and verify the layered structure of the manufactured elements, sclerometer scratch hardness tests were carried out on each sample. Based on the research results, it was possible to observe the porosity growth mechanism and its influence on the material strength (including static and dynamic tests). Parameters modifications that caused 20% lower energy density, as well as elongation of the elements during tensile testing, decreased twice, which was strictly connected with porosity growth. An increase of energy density, by almost three times, caused a significant reduction of force fluctuations differences between both tested surfaces (parallel and perpendicular to the building platform) during sclerometer hardness testing. That kind of phenomenon had been taken into account in the microstructure investigations before and after dynamic testing, where it had been spotted as a positive impact on material deformations based on fused material formation after SLM processing.

## 1. Introduction

Over the last ten years, there has been increasing interest in studies related to metal additive manufacturing [1,2,3,4,5,6,7]. A considerable fraction of these studies are related to powder bed fusion (PBF) techniques, such as selective laser melting (SLM), direct metal laser sintering (DMLS), or electron beam melting (EBM) [8]. The “layer-by-layer” method of part fabrication connected to a powder bed feeding mechanism allows for design freedom and the fabrication of objects with complex shapes that would be impossible to obtain with traditional manufacturing methods [9]. Owing to the specific features of the PBF technique, it is beginning to be used in many demanding branches of industry [10,11,12,13]. However, the growing popularity of SLM, EBM, and DMLS techniques requires detailed knowledge of the technological process, and additional optimization studies are necessary. This problem has been addressed by many researchers [14,15,16,17]. The adopted technological parameters can define the geometric quality of the manufactured objects, as well as mechanical properties, such as the strength, hardness, and roughness of the surface. The optimization of the SLM technological process has been considered in several studies [18,19,20,21,22]. Cherry et al. [18] evaluated the influence of the laser energy density on the properties of 316L stainless steel. The point distance and exposure time were varied, and their impact on the porosity, surface finish, microstructure, density, and hardness were investigated in detail. The results revealed that the laser energy density has an effect on the total porosity. Pleass and Jothi [20] studied the influence of powder characteristics and SLM process parameters on the microstructure and mechanical behavior of Inconel 625. Their results revealed that the size of the applied particle powders affects not only the difficulty of the technological process but also determines the microstructure of the material. Kasperovich and Hausmann [21] focused on a double step optimization procedure for SLM process parameters. They proposed a solution that allows for the improvement of fatigue resistance and ductility in a Ti4Al6V4 titanium alloy. The first optimization step focused on the minimization of inherent defects, while the second step was related to the minimization of internal stresses and adjustment of the microstructure through additional thermomechanical treatment. Larimian et al. [23] studied the effect of scanning speed, scanning strategy, and energy density on the microstructure and mechanical properties by performing microhardness tests, tensile tests, and a scanning microscopy analysis. They found that samples with alternate hatches and a single pass of a laser beam exhibited higher densification and a more refined microstructure. Moreover, higher scanning speeds allowed better mechanical properties to be obtained.

While there are a wide range of technological parameters that determine the mechanical properties and geometric quality of additively manufactured objects, owing to cost and time constraints, the optimization procedure considers only a representative group of parameters [24,25,26]. Some of the most essential issues addressed in these studies are the model orientation in the working chamber [27,28], application of additional heat treatment processes [28,29], and modification of the exposure parameters [24,29,30,31,32,33,34,35,36]. The main purpose of these studies is to determine the influence of each process variable on the mechanical properties and geometric quality of manufactured objects. Furthermore, based on the applied variables, it is possible to not only control the correctness of the manufacturing process but also determine the mechanical properties of the final object. Depending on the adopted process parameters, different material properties can be achieved without any additional post-processing treatment. Furthermore, it is possible to ensure a high geometric accuracy by minimizing the influence of thermal shrinkage effects [37]. These features will have a significant influence on the further development of the additive manufacturing technique and require additional studies.

The main aim of this study was to evaluate the influence of a selected group of SLM technological parameters on the mechanical properties of standard objects made from austenite 316L stainless steel. This material is widely used in the PBF system and has previously been characterized in detail [31,38,39,40,41,42,43]. However, many of the previously reported results are related to studies employing standard technological parameters and examining the influence of these parameters on the resulting mechanical properties in comparison to bulk materials. Precise prediction of the mechanical behavior of objects made through additive manufacturing will require extended studies in which the technological parameters are modified based on the defined methodology. This study is a continuation of the research [17] previously performed at the Military University of Technology. Based on the previously conducted studies, this study will analyze the influence of process energy density on the mechanical behavior of austenite 316L stainless steel. Taking into account the conclusions drawn from previous studies, this study evaluates the modification of the process parameters influence on the porosity of manufactured objects, as well as the surface hardness and strength of the material, subjected to both quasi-static and dynamic loading conditions. Additionally, the material microstructure is also analyzed.

Through these experiments, the influence of three groups of technological parameters on the mechanical behavior of additively manufactured 316L stainless steel was evaluated. The selection of those three parameters groups was strictly connected with use in the preliminary research [17] the energy density values. It had selected the three, most significant parameters groups:
-“S_1” is the reference group, which was defined according to the recommended values given by the SLM device distributors,-“S_17” is characterized by the lowest value of energy density used in preliminary research [17],-“S_30” is based on the best-mentioned parameters in reference [42], which is also characterized by the highest energy density used in the preliminary research [17].

The research methodology used in this study is presented in Figure 1.

## 2. Characteristics of Austenite 316L Material Specimens in the Manufacturing Process

All of the manufactured samples were prepared using the SLM 125HL system (SLM Solutions Group AG, Lubeck, Germany). The specimens used in the tensile tests were defined in accordance with the ASTM E466 96 standard. The major geometric dimensions of the specimens are shown in Figure 2. Additionally, cylindrical bar specimens were also produced (Figure 3) for use in the dynamic compression tests.

### Materials

The metal powder used in this study was manufactured using gas atomization under an argon atmosphere. The manufacturing process was conducted using atomized 316L (1.4404) austenitic steel powder (LPW Technology Ltd., Widnes, UK). The powder grains had a regular spherical shape for the most part; the diameter of the particles was in the range of 15 µm to 45 µm. The density of the material was 7.92 g/cm3, and its flowability was 14.6 s/50 g. Cumulated mass values of the powder particles size distribution has the following values: D10 = 18.22 μm, D50 = 30.50 μm, and D90 = 55.87 μm. Scanning electron microscopy (SEM) images of the powder grains are shown in Figure 4.

The chemical composition of the material is listed in Table 1.

Microstructural analysis of the powder material was performed using an Olympus LEXT OLS 4100 digital light microscope (Olympus Corporation, Tokyo, Japan). For all microscopic analyses, the samples were mounted in resin, ground with 80, 320, 600, 1500, and 2000 grade abrasive papers, and polished using diamond paste (3 μm grade). To reveal the microstructure of the samples, an acetic glycerygia solution (6 mL HCl + 4 mL HNO_3_ + 4 mL CH_3_COOH + 0.2 mL glycerol) was applied with an etching time of 40 s.

Based on previous results published by the authors [17], it was found that modification of selected technological parameters within a low range (± 10%) cause noticeable differences in the mechanical properties of manufactured material specimens. Taking into account the previously obtained results, it was determined that there is a need for further study. Thus, this study focused on minimal and maximal values of the energy density (equation given by Equation (1)) used in preliminary research [17] regarding the default parameters setting (“S_1” parameters group). Three different groups of manufacturing parameters were thus selected, as summarized in Table 2.
(1)$$\rho_{e}=\frac{L_{p}}{e_{v}\cdot h_{d}\cdot l_{t}},$$
where:
L_P_ – laser power [W],e_v_ – exposure velocity [mm/s],h_d_ – hatching distance [mm], andl_t_ – layer thickness [mm].

The technological process parameters subjected to modification were the laser power, exposure velocity, and hatching distance. The choice of this group of parameters is a result of the method for their definition and control. These parameters depend on the applied optical system and energy source. Moreover, the values of the considered parameters can be modified precisely in comparison to the worm gear mechanism responsible for positioning the working platform.

Based on the results of preliminary studies [17], the following groups of technological process parameters were defined:
S_1—default settings recommended for the processing of 316L steel with the SLM 125HL machine;S_17—10% higher exposure speed, 10% higher hatching distance, and 10% lower laser power relative to the reference S_1 group; in a previous study, it has been characterized by the lowest energy density and, as the result, a higher material porosity was obtained under these conditions [17]; S_30—the highest energy density was defined in relation to all of the tested groups of parameters presented in Reference [17] and suggested in Reference [42].

The samples used for the uniaxial quasi-static tensile tests are presented in Figure 5. Additionally, a view of the specimens subjected to quasi-static and dynamic compression tests is shown in Figure 6. All specimens were fabricated under an argon atmosphere with an oxygen content of less than 0.2%.

The tensile samples were oriented horizontally on the building platform to produce the highest possible deformation of the material during tensile testing.

Specimens intended for the compression tests were oriented on the building platform as shown in Figure 6. The proposed specimen orientation was justified by the ability to obtain higher geometrical accuracy. Furthermore, the amount of support structure necessary in this orientation is minimized and will not affect the geometric tolerance of the fabricated specimens.

All of the samples were specifically arranged during the fabrication process in order to obtain the maximal material hardening. This phenomenon is associated with many different material properties along the building direction, where the structure of the material is layered [16]. For the sclerometer hardness, instrumental indenter test, and microstructural analysis, both horizontally and vertically oriented structures were considered.

Geometric models of the manufactured specimens were designed using SolidWorks 2018 software (version 2018). Data preparation and the operating code for the SLM machine were generated in Magics 19 software using the Metal Build Processor module.

## 3. Microstructural Analysis 

Figure 7, Figure 8 and Figure 9 show sample surfaces parallel and perpendicular relative to the machine building platform. The images show that, regardless of the adopted values for the technological parameters, the sides in the perpendicular direction have noticeably higher densification than those in the parallel direction. This phenomenon has also been observed and discussed in previous studies [21]. In all of these figures, the elements porosity is clearly visible, and this feature is directly related to the use of different parameter values [16,22,23]. The porosity is a result of the presence of gas and is caused by the “balling effect” [24] and the presence of non-melted grains in the material volume.

Small pores with an average size of 60 µm are visible in Figure 7, and their presence is associated with thermal shrinkage and balling due to the material solidification [42]. The higher porosity observed in Figure 8 is also associated with an insufficient energy density, which resulted in an increase in the amount of non-melted grains.

Figure 8 also shows pores with different shapes; these are less regular than in the samples of groups S_1 and S_30. This type of porosity may cause stress concentration and be result in the failure of elements during exploitation. The porosity also influences the mechanical properties of materials. This problem was also considered and evaluated in this study. The material microstructure shown in Figure 9 is characterized by good material solidification. It can be seen that the melt pool size in Figure 9a is visually smaller than that in Figure 8a. This is directly related to the manufacturing parameters, as the samples in group S_1 were fabricated with an energy density three times less than that used in the manufacturing process for samples in group S_30. Consequently, the thermal gradient in the S_30 samples was higher and caused rapid solidification, which is connected with a high cooling rate.

The S_30 samples were manufactured using a very short hatching distance (the distance between two consecutive laser exposure lines), which can produce better solidification between the melted tracks (Figure 9a) and layers (Figure 9b) in the surface parallel to the machine building platform. 

## 4. Sclerometer Hardness and Instrumental Indenter Tests 

### 4.1. Testing Method Description

The sclerometer hardness measurement (HSP) involves scratching the surface of an element using an indenter with a specified geometry on Universal Nano & Microtester (UNMT) (CETR, INC; Campbell; USA) e.g., a Rockwell indenter, diamond cutting blades, diamond stylus, etc.); a constant normal load is maintained with an additional constant indenter movement speed. Then, the average width of the scratch is measured (e.g., with a microscope). This method is used to measure the hardness of materials on both macro and micro scales (and, with some modifications, on the nano scale).

The sclerometer hardness test was used to analyze the significance of the layered structure of the material on a cross-section along the Z-axis (the axis perpendicular to the building platform). The cross-sections along the Z-axis were compared to cross-sections made parallel to the building platform, as shown in Figure 10.

### 4.2. Scratching Force Changes and Sclerometer Hardness Analysis

As mentioned in Section 4.1, the sclerometer hardness testing method is based on the measurement of the indentation force during the scratching process. This attribute was useful for analyzing the influence of the layer orientation on changes in the force during the scratching process. The results for each sample are shown in Figure 11.

Figure 11 illustrates that the different force trends and values are not related to the sample properties, but rather to the different shapes of the metallographic section of the sample. For each sample, the machine tools were calibrated to achieve appropriate test values compatible with the testing method standard. The following can be observed:
-for the S_1 parameter group, the force fluctuations on the surface oriented perpendicular to the building platform occur in a small range and are higher than those on the surface oriented parallel to the building platform;-for the S_17 parameter group, the force fluctuations on the surface oriented perpendicular to the building platform are much higher than those on the surface oriented parallel to the building platform;-for the S_30 parameter group, the force fluctuations on the surface oriented perpendicular to the building platform are lower than those on the surface oriented parallel to the building platform.

All of the observed phenomena in three samples are directly related to the influence of the process parameters, especially the energy density. When the energy density decreases, the force fluctuations on the surface oriented perpendicular to the building platform increase. It can be also observed that for the samples manufactured using the S_30 parameter group, the fluctuations the on surface oriented perpendicular to the building platform were lower than those on the surface oriented parallel to the building platform. This phenomenon is related to the very high amount of energy delivered in this case, which caused a greater range of material solidification.

The indentation force measurement during the tests was considered because it cannot be analyzed based on sclerometer hardness results, which compare the scratch dimensions with the indentation force. The dimensions of the scratches on the element surfaces of the elements are in opposition to the results for the indentation force as a function of scratch length, as shown in Table 3.

The results of the sclerometer hardness tests reveal that a higher energy density results in a lower sclerometer hardness value. Furthermore, for the specimens manufactured using the S_1 and S_17 technological process parameter values, the sclerometer hardness on the surface of the layers (oriented parallel to the building platform) was higher than that on the surface passing through the layers (oriented perpendicular to the building platform). The deviation from this result in sample S_30 is a consequence of using a very high energy density on the layers and inducing material strengthening in this direction in order to obtain a high range of material fusion.

### 4.3. Instrumental Hardness Analysis

The instrumental hardness testing method was used to analyze the impact of the indenter on the material surface by measuring both the force and displacement (depth) during plastic and elastic deformation. For these measurements, only the surface oriented perpendicular to the building platform was considered. An entire force loading and unloading cycles was recorded for further analysis, making it possible to generate values equivalent to the traditionally measured hardness. The results of the measured instrumental hardness are shown in Table 4.

The instrumental hardness results do not reveal any relationship between the hardness and the parameter modifications. A very similar phenomenon was observed during micro-hardness testing in earlier research [16]. It can be assumed that hardness testing methods, such as instrumental testing and traditional micro-hardness testing, are only appropriate to define the material properties of materials that are more isotropic than additively manufactured elements. As noted in Section 4.2 of this study, the sclerometer method is more appropriate for additively manufactured elements because it is possible to observe even very small differences in the material properties.

## 5. Tensile Tests with Digital Image Correlation (DIC) Analysis

Axial tensile strength tests of SLM additively manufactured samples made of 316L steel were carried out according to the ISO standard [44] with the use of the hydraulic pulsator, Instron 8802 (Instron, Norwood, MA, USA). Measurements of the deformation under axial stretching were obtained using an Instron 2630-112 extensometer with a measuring base of 50 mm. All samples subjected to axial tension had the same geometry. The results of monotonic tensile tests on samples produced by the SLM technology with 316L steel are shown in Figure 12.

The orientation of the laser connection has a significant impact on the strength properties of the additively manufactured elements. For sample S_1, the ultimate tensile strength was 666 MPa, and its elongation was 44.91%. The ultimate tensile strength for the S_30 specimens was 631 MPa, which represents a decrease in strength of approximately 5.3%. For the elongation, a slight 1% increase to 45.88% was observed. For specimens in group S_17, the breaking strength was 557 MPa and elongation was 27.21%. Compared to case S_1, the S_17 breaking strength decreased by approximately 14.4%, while the elongation decreased by 17.7%. Analysis of the sample surface deformation process during the monotonic tensile tests was carried out using the digital image correlation (DIC) method. Observations of the deformation were carried out using Dantec Q-400 system (DANTEC DYNAMICS A/S; Skovlunde; Denmark) for three different specimen series (S_01, S_17, and S_30) considering three characteristic parameters: the yield strength, ultimate tensile strength, and breaking strength. Received data from DIC system and tensile test machine was evaluated by ISTRA 4D software. The results of these tests are shown in Figure 13.

The test results (Figure 13) for the points marked with numbers 1 and 2 in Figure 14 show a homogeneous distribution of deformation over the entire surface for all three samples. A visible difference can be observed in the S_17 sample results; after exceeding the yield point (Figure 13b), areas of uneven material deformation can be observed. It should be noted that the deformation images do not show banding. That type of phenomenon can affect the material structure and residual stresses caused by the manufacturing process, which can have an influence on the strength properties [41]. For samples S_1 and S_30 (Figure 13c), the maximum concentration of deformations was observed at the crack initiation site. The maximum deformation location and the continuous nature of their growth indicates that the material is consistent throughout the load range of each sample. In the area with high deformation (Figure 13c), a narrowing of the material is visible, which indicates the high plasticity of the produced material. In the case of sample S_17, after exceeding the conventional limit of the immediate strength, areas of banded heterogeneity in the material deformation were observed. This could be attributed to the presence of residual stress in the material [41]. During the tests, there were no visible phenomena to indicate the crack initiation locations.

## 6. Static and Dynamic Compressive Tests 

A subsequent stage of investigations was conducted to determine the influence of the varied groups of technological parameters on the material strength. The mechanical response of the material was evaluated under quasi-static and dynamic loading conditions. Uniaxial compression tests were carried on cylindrical bar samples with a diameter Ø = 6.0 mm and length L = 6.0 mm. These samples were manufactured additively with the three groups of selected SLM technological parameters. 

The first stage of the strength experiments was performed using an MTS Criterion 45 universal strength machine (MTS Systems Corporation; Eden Prairie; MN, USA). A view of the machine and typical compression test results is presented in Figure 14. Based on the obtained uniaxial stress–strain plots, it could be determined that the groups of technological parameters affect the material strength properties. The highest strength was obtained when the reference (S_1) group of parameters was used. Specimens manufactured using the S_17 and S_30 parameter groups demonstrated lower mechanical strengths. The results obtained for the S_30 group of technological parameters are particularly interesting. Despite application of a higher energy density, no increase in the mechanical strength was observed. This result is contrary to those that have been presented previously [42].

The other stage of the mechanical characterization was the high-strain-rate uniaxial compression tests, which were carried out using a Split–Hopkinson pressure bar (SHPB) laboratory setup. Dynamic compression tests were performed with strain rates varying from 1170 s−1 to 2035 s−1 using specimens like those for the uniaxial quasi-static tests. Typical results are presented in Figure 16.

Comparing the quasi-static and dynamic compression results reveals that, regardless of the group of technological parameters used, the fabricated material samples exhibit a strain rate sensitivity (Figure 15, Figure 16, Figure 17 and Figure 18).

To characterize the phenomenon of material plastic deformation caused by the Split–Hopkinson dynamic tests, the microstructures of the samples were analyzed after the tests and compared with the observed microstructures before the test. A comparison of the microstructures of each sample group is shown in Figure 19.

After comparison of the microstructure and compression test results, some similarities can be noted. The most significant melt pool deformation can be observed in the sample manufactured using the S_17 parameters group, which also had the lowest instrumental hardness of the measured groups. The worst mechanical strength properties of all three groups could be attributed to the high porosity of the S_17 samples [17]. There is also a visible difference between the microstructure images for the S_01 and S_30 specimens. The deformation of melt pools in the S_01 sample are more regular than in the S_30 sample. The presence of cracks between the material layers after the dynamic tests can also be observed, and it can be inferred that some amount of energy was dissipated during propagation of these cracks between layers.

## 7. Conclusions

The technological parameters of the SLM process strongly affect the mechanical properties and microstructure of additively manufactured elements. This is generally caused by thermodynamic phenomena that have a significant influence on the specimen fabrication process. Based on the results obtained in this study, it was found that even low-range modification of the standard values of technological parameters will have an influence on the mechanical properties of the material and result in different outcomes. Furthermore, application of the highest energy density did not produce an improvement in the material mechanical behavior. Based on the results, the following conclusions were formulated:
Changing the exposure parameters (laser power, exposure speed, and hatching distance) affects the melt pool size and porosity of the structure of the produced material and consequently affects the mechanical properties of additively manufactured elements composed of 316L austenitic steel. All mentioned parameters significantly affect the process energy density which is introduced into a distributed powder layer. The value of the energy density directly affects the melting pool temperature, its size, and, at the end, all thermal history of the produced elements. All these factors significantly affect the mechanical properties, which was proven in this research paper.The results of microstructural analyses of the manufactured material before and after dynamic tests allow for an initial assessment of its static strength. The highest level of compression stress of S_1 samples is covered by the highest tensile strength of those samples. It was also registered in DIC images, where the area of high material deformations is much bigger than in the other, tested samples.Analysis of the microstructural and strength test results demonstrates the possibility of designing the fabricated material for specific applications. One of the most important examples could be porosity growth with decreasing the process of energy density. Forced porosity growth could be helpful in high-porous friction bearing or medical applications where it is necessary to assure tissue deposition in produced prostheses.The energy dissipation capacity of the resulting structures affects the observed mechanism of material cracking during dynamic loading, particularly in the areas on the border between adjacent layers. Registered different material behavior is strictly connected with its properties. These properties are connected with porosity types in which generation was caused by different process parameters. After dynamic tests, different material behavior was visible, wherein S_1 samples the deformation process was stable in the whole analyzed area. S_17 samples were characterized by a high amount of cracks, which were generated near non-fused powder grains. Different cracking behavior was presented in S_30 samples images, where cracks were mostly generated between melt pools. That type of phenomenon is connected with too high energy density, which directly influences the “key-hole” porosity generation in connections between melt pools.

All of the analyses presented in this paper are beneficial for understanding the influence of different technological process parameters on the mechanical properties of fabricated elements. Based on the obtained results, it could be concluded that modification of the technological process parameters can be utilized to obtain some specific material features, as well as allows for materials behavior during loading prediction that could be essential, depending on the material application.

## Figures and Tables

**Figure 1 materials-13-01449-f001:**
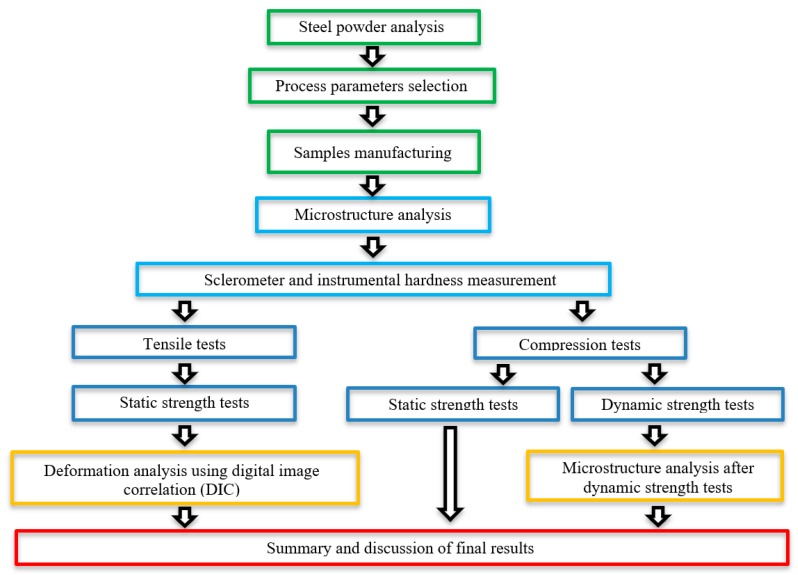
Illustration of the research methodology employed in this study.

**Figure 2 materials-13-01449-f002:**
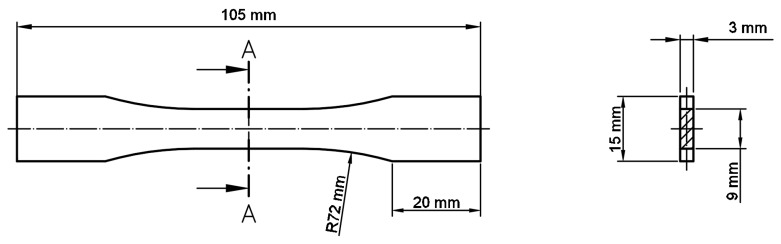
Dimensions of the ASTM E466 96 standard tensile test dog-bone sample.

**Figure 3 materials-13-01449-f003:**
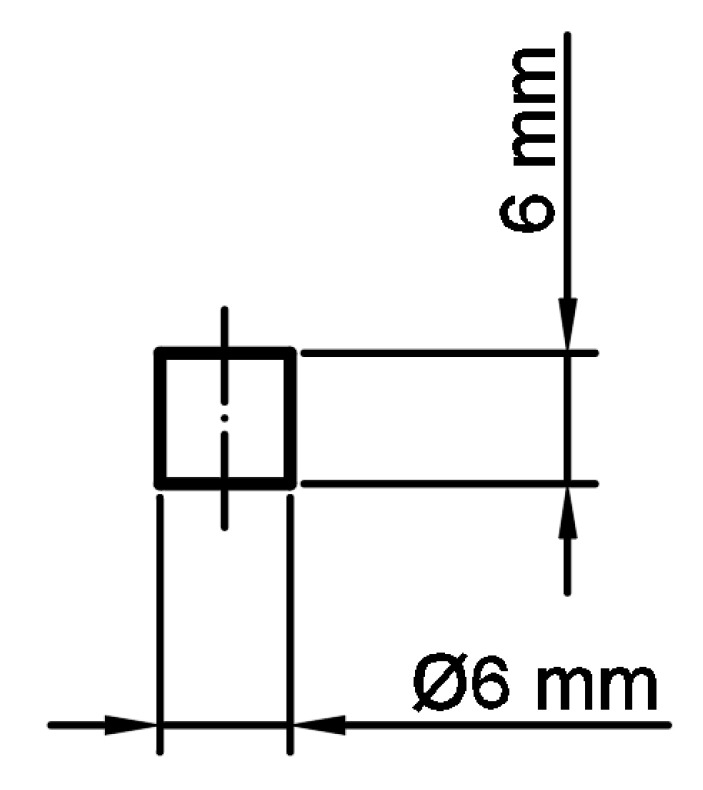
View of cylindrical bar specimens subjected to dynamic compression tests.

**Figure 4 materials-13-01449-f004:**
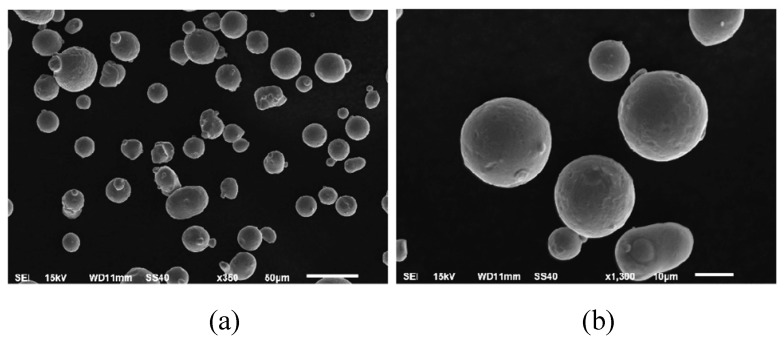
SEM images of 316L powder grains captured at scales of (**a**) 50 µm and (**b**) 10 µm.

**Figure 5 materials-13-01449-f005:**
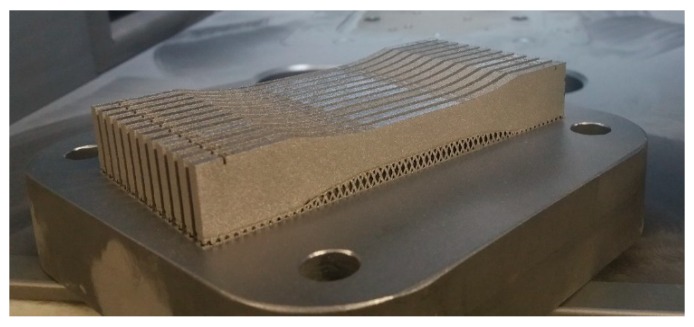
Tensile test samples on the selective laser melting (SLM) 125 HL machine building platform.

**Figure 6 materials-13-01449-f006:**
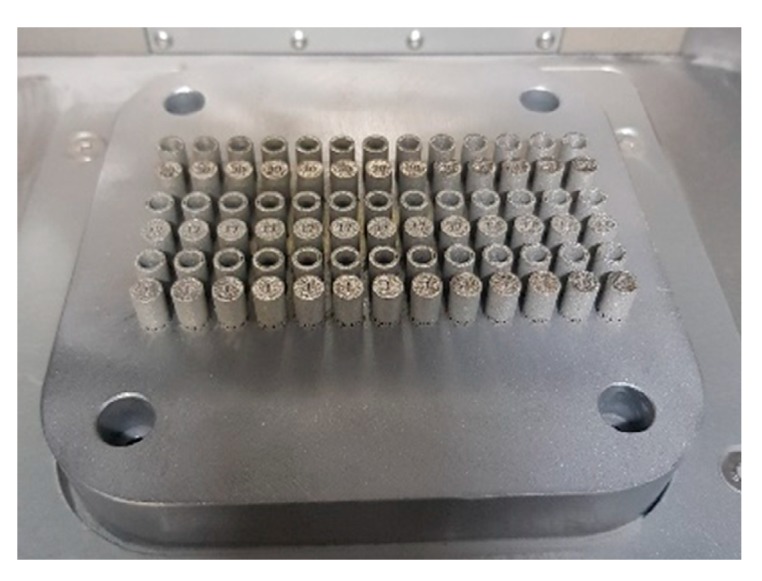
Samples for the static and dynamic compressive tests on the SLM 125 HL machine building platform.

**Figure 7 materials-13-01449-f007:**
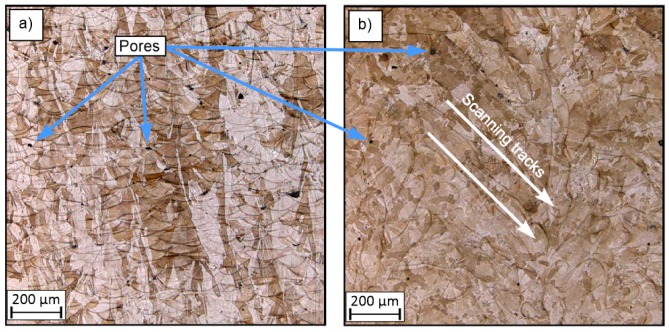
Microstructure of the surfaces of sample S_1: (**a**) perpendicular to the machine building platform and (**b**) parallel to the machine building platform.

**Figure 8 materials-13-01449-f008:**
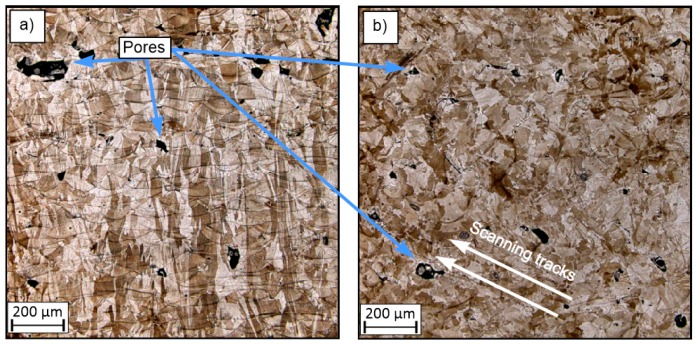
Microstructure of the surfaces of sample S_17: (**a**) perpendicular to the machine building platform and (**b**) parallel to the machine building platform.

**Figure 9 materials-13-01449-f009:**
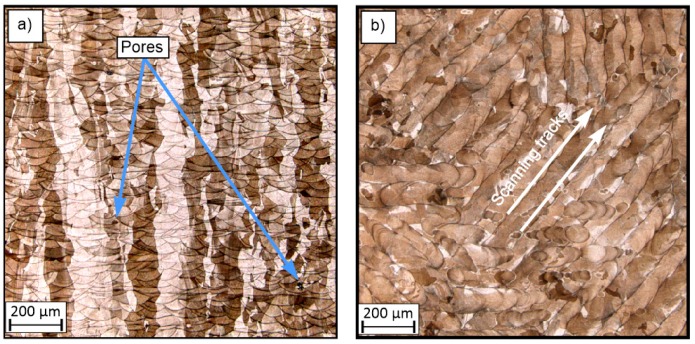
Microstructure of the surfaces of sample S_30: (**a**) perpendicular to the machine building platform, and (**b**) parallel to the machine building platform.

**Figure 10 materials-13-01449-f010:**
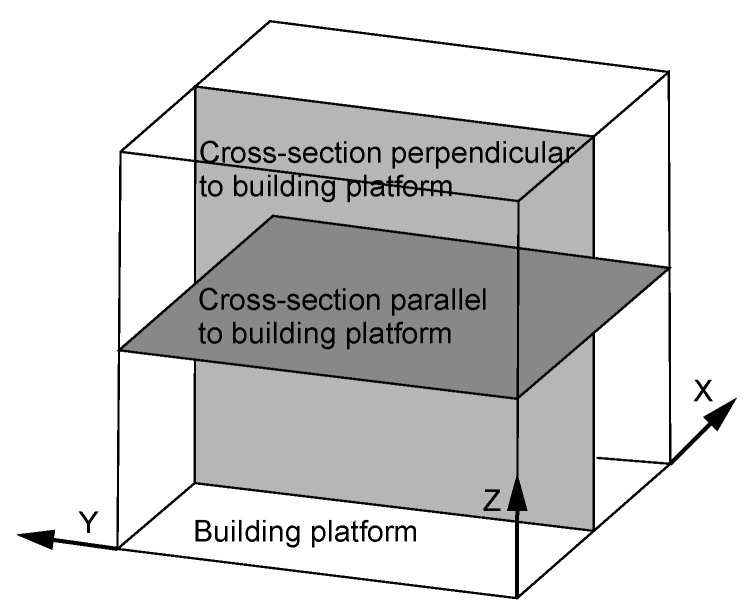
View of the element characteristic zones.

**Figure 11 materials-13-01449-f011:**
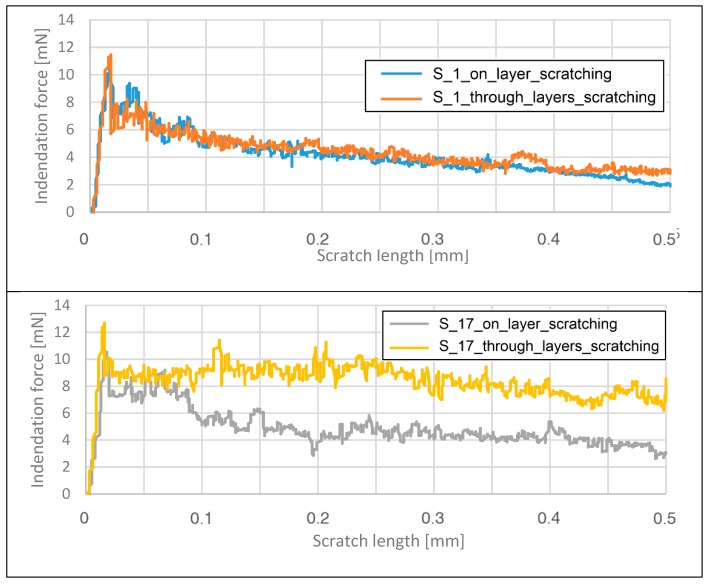
Variation in the indentation force as a function of the scratch length.

**Figure 12 materials-13-01449-f012:**
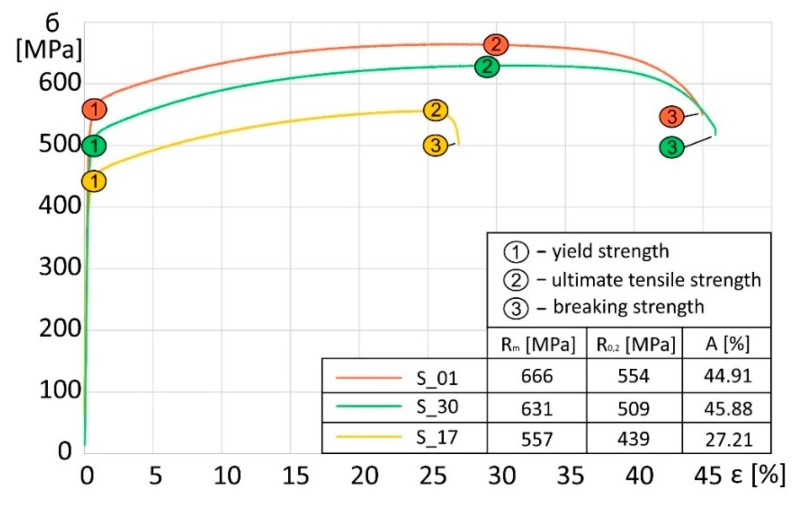
Monotonic stretching curves for S_1, S_17, and S_30 samples.

**Figure 13 materials-13-01449-f013:**
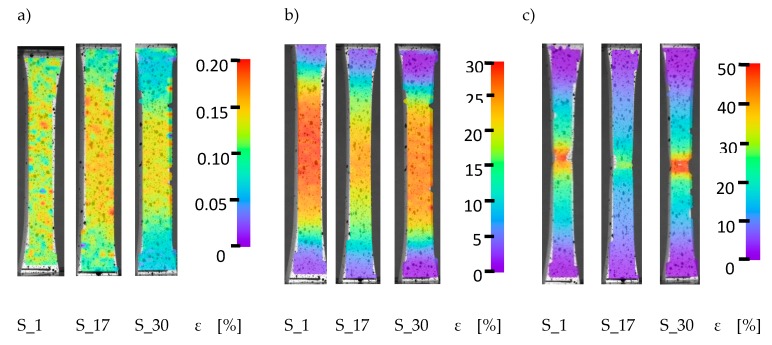
Strain distribution during monotonic tensile tests: (**a**) yield strength, (**b**) ultimate tensile strength, and (**c**) breaking strength.

**Figure 14 materials-13-01449-f014:**
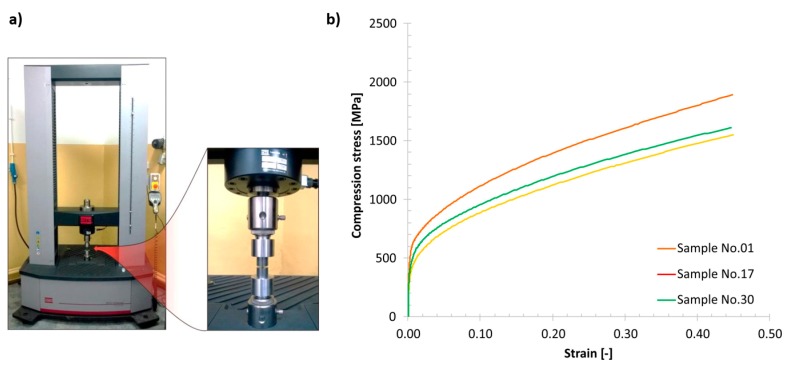
(**a**) Universal strength machine with a detailed view of the specimen mounting assembly, and (**b**) compression plots obtained for various 316L material samples.

**Figure 15 materials-13-01449-f015:**
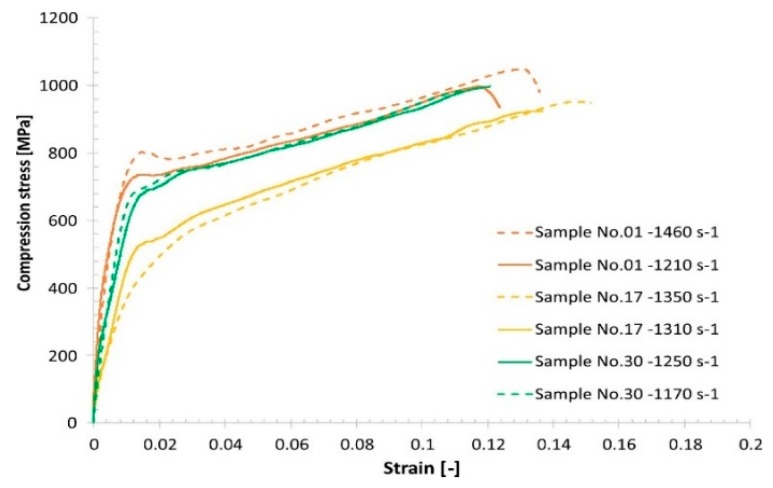
Results of the Split–Hopkinson pressure bar (SHPB) compression tests under dynamic loading conditions.

**Figure 16 materials-13-01449-f016:**
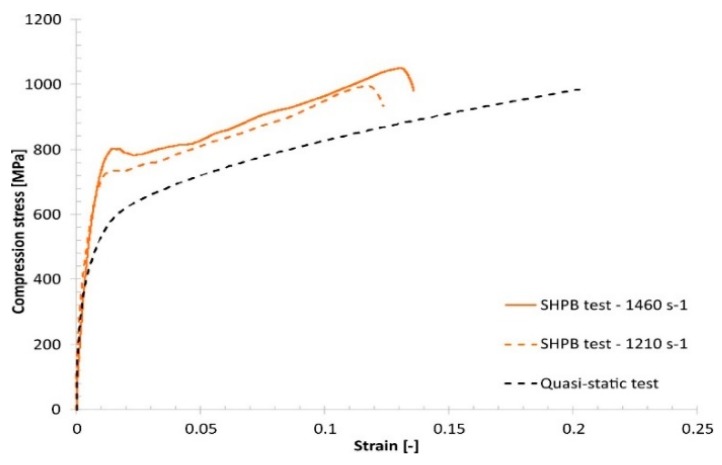
Comparison of the dynamic and quasi-static uniaxial compression test results obtained for specimen No. 01.

**Figure 17 materials-13-01449-f017:**
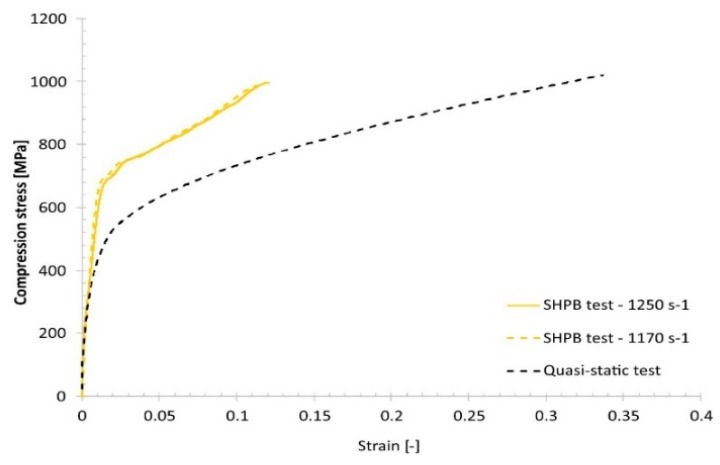
Comparison of the dynamic and quasi-static uniaxial compression test results obtained for specimen No. 17.

**Figure 18 materials-13-01449-f018:**
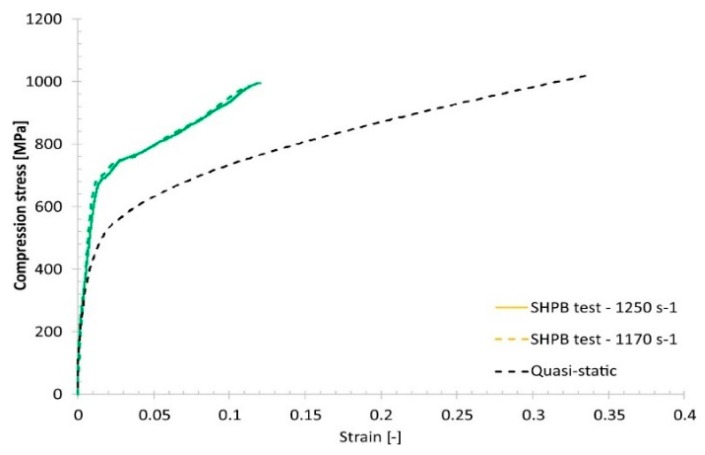
Comparison of the dynamic and quasi-static uniaxial compression test results obtained for specimen No. 30.

**Figure 19 materials-13-01449-f019:**
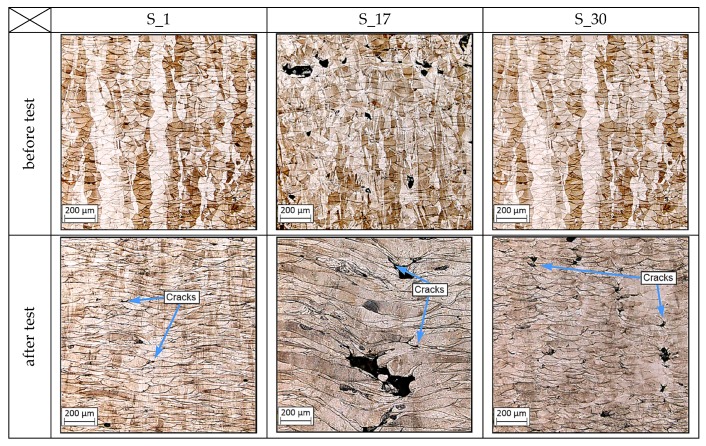
Microstructures of the samples before and after the Split–Hopkinson tests.

**Table 1 materials-13-01449-t001:** Chemical composition of 316L austenitic steel.

C	Mn	Si	P	S	N	Cr	Mo	Ni
**weight [%]**								
**max. 0.03**	max. 2.00	max. 0.75	max. 0.04	max. 0.03	max. 0.10	16.00-18.00	2.00- 3.00	10.00-14.00

**Table 2 materials-13-01449-t002:** Specifications of selected groups of technological process parameters.

Parameter Sets	Laser Power, L_P_ [W]	Exposure Velocity, e_v_ [mm/s]	Hatching Distance, h_d_ [mm]	Energy Density, ρ_E_ [J/mm^3^]
S_1	190	900	0.12	58.64
S_17	180	990	0.13	46.62
S_30	120	300	0.08	166.67

**Table 3 materials-13-01449-t003:** Sclerometer hardness of samples S_1, S_17, and S_30 in both measured planes.

Sample Description	Sclerometer Hardness [GPa]
S_1—measurement in parallel surface (regarding to build platform)	4.16
S_1—measurement in perpendicular surface (regarding to build platform)	4.44
S_17—measurement in parallel surface (regarding to build platform)	3.67
S_17—measurement in perpendicular surface (regarding to build platform)	4.04
S_30—measurement in parallel surface (regarding to build platform)	3.85
S_30—measurement in perpendicular surface (regarding to build platform)	3.53

**Table 4 materials-13-01449-t004:** Instrumental hardness of samples S_1, S_17, and S_30.

Sample Description	Instrumental Hardness [GPa]
S_1—measurement in parallel surface (regarding to build platform)	1.74
S_17—measurement in parallel surface (regarding to build platform)	1.31
S_30—measurement in parallel surface (regarding to build platform)	1.37

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
