# Peer review of "Influence of Selective Laser Melting Technological Parameters on the Mechanical Properties of Additively Manufactured Elements Using 316L Austenitic Steel"

_materials, 2020, doi:10.3390/ma13061449_

Round 1

Reviewer 1 Report

The papers starts OK, the introduction is well written. But the experimental part and the results need to be improved.

All of the micrographs that show the microstructure are awful. The scales are practically invisible, the microstructure is blurry, and the blue arrows that should point to the pores in figs 7-9 are pointed at nothing. Please make the figurtes readable and increase the quallity.

The characterisation of the powder that was used in the experiment is just a SEM micrograph with no real explanation or analysis.

The micrographs are not good, but I am afraid the authors are confusing "melt pool residue" and austenitic grains, they are not the same. And they mention grain size but give no figures, please estimate the grain size, I know it can not be done in the usual way because the grains are elongated, but give something more concrete if you mention it.

The graph in figure 11 is chaotic and simply looks unprofessional, fix the mess.

Figures 12 and 13 are just boring and uninformative graphs. Why not make them a informative tables instead? It would save space and be more clear.

The sample names S_1_tl and so on look like something from a computer code and not an actual scientific paper.

The importance of the SHPB dynamic test is hidden please elaborate why you did the experiments and why they are important or useful. The whole section from figure 18 to 21 is there for no apparent reason.

Again figure 22 improve the metallographic images and description of the images. 

Also figure 17 belongs in the experimental part.

Figure 16 is difficult to read.

The conclusion 2 is missing a lot of results and is not really supported by the paper data.

I think that the idea behind the paper is good, and that the paper has good potential, but it needs to be significantly improved in the metallographic department and in the elaboration of the results of the dynamic tests.

Author Response

Response to the Reviewer's comments

Dear Reviewer, 

We would like to thank you for all your comments.

According to your remarks, the following issues had been taken into account:

(Please be informed that we used a yellow highlighter to point changes in our paper made based on your comments. The green highlighted parts are based on the other Reviewer comments)

  1. All of the micrographs that show the microstructure are awful. The scales are practically invisible, the microstructure is blurry, and the blue arrows that should point to the pores in figs 7-9 are pointed at nothing. Please make the figurtes readable and increase the quallity.

Ad.1. We used additional software to make elements structure more visible and to create new scales to make it more visible. Additionally, we put new arrows and marks in the graphical software to avoid deformations made by MS Word software. Finally, we exported the microstructure photos to not lost the quality.

  1. The characterisation of the powder that was used in the experiment is just a SEM micrograph with no real explanation or analysis.

Ad. 2. For the powder, we put a standard description based on own SEM analysis. The powder had been provided by the professional manufacturer of that kind of materials and as it is typically done we have received a quality test certificate. Anyway, we put some additional data about the powder – it is yellow highlighted: “The density of the material was 7.92 g/cm3 and its flowability was 14.6 s/50 g. Cumulated mass values of the powder particles size distribution has the following values: D10 = 18.22 μm, D50 = 30.50 μm, D90 = 55.87 μm.”

  1. The micrographs are not good, but I am afraid the authors are confusing "melt pool residue" and austenitic grains, they are not the same. And they mention grain size but give no figures, please estimate the grain size, I know it can not be done in the usual way because the grains are elongated, but give something more concrete if you mention it.

Ad. 3. Indeed it is better to use melt pool terminology and it is a better idea. We were not intending perception melted pools as austenitic grains. Regarding your comments, we changed all “grain words” to “melt pool”. According to estimating the melt pools size, as you mentioned it cannot be properly estimated and could be verified based on microstructural observations. We tried to make some measurements but it could be interpreted as biased. In that case, we have changed the sentence and now it is so: “It can be seen that the melt pool size in Figure 9a is visually smaller than that in Figure 8a.”

  1. The graph in figure 11 is chaotic and simply looks unprofessional, fix the mess.

Ad.4. We divided one graph into three separate and put it in one figure.

  1. Figures 12 and 13 are just boring and uninformative graphs. Why not make them a informative tables instead? It would save space and be more clear.

Ad. 5. Regarding your comments, we changed the mentioned figures into informative tables.

  1. The sample names S_1_tl and so on look like something from a computer code and not an actual scientific paper.

Ad. 6. During changing the mentioned figures into tables we have changed also samples names

  1. The importance of the SHPB dynamic test is hidden please elaborate why you did the experiments and why they are important or useful. The whole section from figure 18 to 21 is there for no apparent reason.

Ad. 7. The SHPB studies enable the determination of the mechanical response of the material under dynamic loading conditions. This type of investigation allows for the evaluation of the material strain rate sensitivity. This information is crucial especially when a material is planned to be used in an application where impact loading conditions exist. Presented in Fig.18-21 plots demonstrate the difference between results of compression tests obtained under quasi-static and impact loading conditions. Based on them it can be observed the difference of mechanical response of 316L stainless. Referring to the number of figures demonstrating the gathered results, the Authors’ decided to present them in a separate form due to better clarity. Analyzing these plots it can be seen the visible sensitivity of 316L  stainless steel, 3D printed material under dynamic loading conditions.

  1. Again figure 22 improve the metallographic images and description of the images.

Ad. 8. We have made similar changes as in the before-mentioned metallographic images.

  1. Also figure 17 belongs in the experimental part.

Ad. 9. To be consistent in “not showing the testing stands” we removed this figure to make possible to zoom the SHPB pictures and reduce a little a big volume of our paper.

  1. Figure 16 is difficult to read.

Ad. 10. It was zoomed

  1. The conclusion 2 is missing a lot of results and is not really supported by the paper data.

Ad. 11. We extended the description by covering the statement using results from tensile tests and digital image correlation analysis. It was yellow-highlighted in the text.

Once again we would like to thank you for your kind and valuable comments. We hope it meets your expectations.

Sincerely,

Janusz Kluczyński

Military University of Technology

Faculty of Mechanical Engineering,

Institute of Robots & Machine Design

2 Gen. S. Kaliskiego Street, 00-908 Warsaw 49, Poland

Tel.: +48 261 837 208

Fax: +48 261 837 366

[email protected]

Reviewer 2 Report

General comment:

I congratulate the authors for the well-structured paper on the Selective Laser Melting 316L austenitic steel. The paper may be ACCEPTABLE for publishing in Materials after a manuscript MINOR REVISION. My specific comments are given below.

Specific comments:

Comment 1: In my opinion this study is relevant but it is not an optimization study. Ref. 16 (from the group) is an optimization study.

Comment 2: In other words, the key issue of this study is that authors were not able to demonstrate a logic/scientific “bridge“ between the previously published that (Ref. 16) and the present one.

 Comment 3: The explanation paragraph on page 5 is not well explained and sufficient.

Comment 4: Authors wrote: “The group of process parameters 20 considered was selected from the first-stage parameters identified in preliminary research”. Why didn’t authors include more conditions in the present paper? Is it possible? Don’t authors think that in SLM parameters optimization studies it is very important to have a considerable number of conditions?

Comment 5: The author should revise the entire manuscript, particularly, the abstract and introduction and expose the study in a different way.

Comment 6: The authors could discuss deeply the overall results.

Comment 7: The conclusions are too vague.

I hope that the Authors will consider my comments helpful in terms of improving their work. I wish all the best in further scientific work to all the Authors.

Author Response

Response to the Reviewer's comments

Dear Reviewer, 

We would like to thank you for all your valuable comments.

According to your remarks, the following issues had been taken into account:

(Please be informed that we used a green highlighter to point changes in our paper made based on your comments. The yellow highlighted parts are based on the other Reviewer comments)

  1. In my opinion this study is relevant but it is not an optimization study. Ref. 16 (from the group) is an optimization study.

Ad.1. We changed the statement about optimization on: “The main aim of this study is to investigate the influence of different energy density values used for the additively manufactured elements using selective laser melting (SLM).” It was green-highlighted in the text.

  1. In other words, the key issue of this study is that authors were not able to demonstrate a logic/scientific “bridge“ between the previously published that (Ref. 16) and the present one.

Ad.2. We would like to thank you very much for this comment. Indeed, in the text, there is a lack of the basis of those three parameters groups selection. The main reason for this selection was the energy density value. We changed the last paragraph of the introduction part and green-highlighted it. That is how it is now:

“ Through these experiments, the influence of three groups of technological parameters on the mechanical behavior of additively manufactured 316L stainless steel was evaluated. The selection of those three parameters groups was strictly connected with used in the own preliminary research [17] the energy density values. It had been selected three, most significant parameters groups:

- “S_1”, is the reference group, which was defined according to the recommended values given by the SLM device distributors,

- “S_17”, is characterized by the lowest value of energy density used in preliminary research [17],

- “S_30”, is based on the best-mentioned parameters in reference [42], which is also characterized by the highest energy density used in the preliminary research [17].”

  1. The explanation paragraph on page 5 is not well explained and sufficient.

Ad.3. We have extended the explanation paragraph with regards to minimal and maximal energy density results – it has been green-highlighted

  1. Authors wrote: “The group of process parameters 20 considered was selected from the first-stage parameters identified in preliminary research”. Why didn’t authors include more conditions in the present paper? Is it possible? Don’t authors think that in SLM parameters optimization studies it is very important to have a considerable number of conditions?

Ad.4. We were considering using a higher amount of conditions but to reasons were appeared during the paper topic preparation:

  • The selection of a higher amount of parameters would significantly increase the paper volume which is now quite big.
  • During the tests, it had been used three different parameters – characterized by the highest and lowest energy density used in the own preliminary research (S_17 and S_30). Those two groups were compared to the default parameters group (S_1). Using some other parameters from the preliminary research would result in the placement of additionally made samples “just between” analyzed cases. It would not provide any additional data and also would make the results vaguer.

  1. The author should revise the entire manuscript, particularly, the abstract and introduction and expose the study in a different way.

Ad.5. Basing on your comments from earlier points we changed the meaning of the introduction and experimental description by referring to extreme energy density values which were compared to the results of the elements manufactured using. We have removed all non-clear data about some kind of process optimization.

  1. The authors could discuss deeply the overall results.

Ad. 6. We extended all results by using wide characteristics and descriptions of reported phenomena. All new elements were green-highlighted.

  1. The conclusions are too vague.

Ad.7. We hope that extending our conclusions basing on your No. 6 comment made conclusions more clear and understandable.

Once again we would like to thank you for your kind and valuable comments. We hope it meets your expectations.

Sincerely,

Janusz Kluczyński

Military University of Technology

Faculty of Mechanical Engineering,

Institute of Robots & Machine Design

2 Gen. S. Kaliskiego Street, 00-908 Warsaw 49, Poland

Tel.: +48 261 837 208

Fax: +48 261 837 366

[email protected]

Round 2

Reviewer 1 Report

The authors have done a very good job with the improvements.The article is ready for publication. I just have one more issue with the statement: In all samples, the fractures were brittle. This is not supported by evidence also the elongations are high for a brittle material, although the normal values for austenitic SS are higher. I think the most simple solution is to remove the sentence. 

Author Response

Dear Reviewer, 

Thank you very much for your suggestion. We greatly appreciate your help in the improvement of our research paper. Regarding your advice we have already removed the mentioned sentence. You can find corrected version in attachment

Sincerely,

Janusz Kluczyński

Military University of Technology

Faculty of Mechanical Engineering,

Institute of Robots & Machine Design

2 Gen. S. Kaliskiego Street, 00-908 Warsaw 49, Poland

Tel.: +48 261 837 208

Fax: +48 261 837 366

[email protected]
